# Explainable one-class feature extraction by adaptive resonance for anomaly detection in quality assurance

**Hootan Kamran**[1], **Dionne Aleman**[1]*, **Chris McIntosh**[2], **Tom Purdie**[2]

**1** Department of Mechanical and Industrial Engineering, University of Toronto, Toronto, ONT, Canada,
**2** Department of Medical Biophysics, University of Toronto, Toronto, ONT, Canada

\* aleman@mie.utoronto.ca

## Abstract

In this study, we address the inherent challenges in radiotherapy (RT) plan quality assessment (QA). RT, a prevalent cancer treatment, utilizes high-energy beams to target tumors while sparing adjacent healthy tissues. Typically, an RT plan is refined through several QA cycles by experts to ensure it meets clinical and operational objectives before being considered safe for patient treatment. This iterative process tends to eliminate unacceptable plans, creating a significant class imbalance problem for machine learning efforts aimed at automating the classification of RT plans as either acceptable or not. The complexity of RT treatment plans, coupled with the aforementioned class imbalance issue, introduces a generalization problem that significantly hinders the efficacy of traditional binary classification approaches. We introduce a novel one-class classification framework, using an adaptive neural network architecture, that outperforms both traditional binary and standard one-class classification methods in this imbalanced and complex context, despite the inherent disadvantage of not learning from unacceptable plans. Unlike its predecessors, our method enhances anomaly detection for RT plan QA without compromising on interpretability—a critical feature in healthcare applications, where understanding and trust in automated decisions are paramount. By offering clear insights into decision-making processes, our method allows healthcare professionals to quickly identify and address specific deficiencies in RT plans deemed unacceptable, thereby streamlining the QA process and enhancing patient care efficiency and safety.

## Author summary

Neural Networks, Adaptive Resonance Theory, Anomaly Detection, Outlier Detection, One-Class Classification, Quality Assurance, Explainable, Interpretable

**Data availability statement:** Data cannot be shared publicly. We use two clinically curated radiotherapy datasets of breast and prostate cancer patients. The datasets were obtained

over three years from Princess Margaret Cancer Centre (Toronto, Canada) under an institutional ethics-approved research protocol. We obtained Research Ethics Board (REB) approval from the University of Toronto. The protocol reference numbers are 12-5225, and 14-8165.

**Funding:** No authors have competing interests.

**Competing interests:** The author(s) received no specific funding for this work.

# 1 Introduction

An estimated 225,800 new cancer cases and 83,300 cancer deaths occurred in Canada in 2020, about 40% of which will be treated with external beam radiation therapy [1]. RT treatment plans must undergo a comprehensive quality assurance (QA) process that requires highly-skilled interdisciplinary man-hours. Machine learning algorithms can help automate QA by analyzing features that led to the creation of acceptable and unacceptable plans in the past [2]. However, a comprehensive RT QA requires machine learning on high-dimensional datasets that store information about the patient's anatomy, the dose volumes, and the machine deliverability. On the other hand, real clinical datasets have a small number of plans (because of inter-institutional variabilities [3] and changing treatment guidelines even within institutions), and are label-imbalanced in favor of acceptable plans (because unacceptable plans, even if recorded, may be overridden by improved plans). We therefore treat RT QA as an anomaly detection problem on small-sample high-dimensional datasets where anomalies are unacceptable. We developed a one-class classifier (OCC) based on adaptive resonance theory (ART), which we call OART, to detect unacceptable treatment plans. OART selects stable features and extracts their memory values to improve anomaly detection by dimensionality reduction, and allows for direct interpretation of classification results in terms of the input features.

The quality of a plan may be unacceptable for a variety of reasons: poorly delineated regions of interest (ROIs) may lead to unrealistic estimations of the dosimetric variables [4], poorly designed delivery plans may lead to delivery complexities [5], and inaccurately-estimated dose volumes may lead to under/overdosages of the tumour(s)/organs at risk (OARs) [6], making traditional binary classification of acceptable v. unacceptable plans challenging, especially as there is a risk that a future unacceptable plan is a truly novel anomaly in that it does not look like historical unacceptable plans. Prior work demonstrated the feasibility of binary classification for RT QA [2], but these approaches rely on the assumption that all error types have been observed in training. This limitation motivates the use of anomaly detection methods, such as our OART model, which learns decision boundaries from only acceptable plans and generalizes to novel errors. We therefore develop a one-class framework for RT QA where norms are learned by training on data from only one class (the majority; here, the acceptable plans) and in testing, objects that deviate from the learned norms are considered unacceptable.

To measure adherence to immediate clinical standards, anatomical [7], dosimetric [8], and machine deliverability [9] features must all be present in the training data. Handling the sparsity in high-dimensional feature spaces is crucial, as highlighted in previous studies on RT planning [10]. Our approach, OART, explicitly addresses this challenge by incorporating a feature selection mechanism that stabilizes anomaly detection results while maintaining interpretability. In such problems as RT planning, where datasets have small sample sizes, high-dimensional feature spaces introduce a particularly high overfitting risk, especially of the underrepresented class [49], in this case, unacceptable plans. We additionally perform binary classification as a baseline to evaluate performance against OCC using our OART algorithm.

## 1.1 Binary classification

To compare or baseline against OCC, we use the following binary classification algorithms: logistic regression, random forests, support vector machines (SVM), naive Bayes, and multilayer perceptron artificial neural networks (NN). These methods are briefly described here.

The binary classification problem takes a set of $m$ inputs, each with $n$ real-valued features, and one binary label. Given $X \in \mathbb{R}^{m \times n}$, $Y \in \{0, 1\}^{m \times 1}$, find $f^{\theta}$ such that $f^{\theta}(X)$ approximates $Y$ as well as possible, the error is called loss. The idea is that, a trained function $f^{\theta^*}(X)$ that achieved low loss may then, by induction, be used on a new input, say, $X'$ to predict their unknown labels $Y'$. Depending on the algorithm, the function $f^{\theta}$ may be a logistic function applied to a weighted sum (logistic regression), stacked nonlinearities (random forests, neural nets), a hyperplane (SVM), or class-conditional probabilities (NB).

## 1.2 One-class classification

One common approach for anomaly detection is with one-class classification [11], where only acceptable samples are available in training. Unacceptable samples are included in the testing set for performance metrics. There are algorithms for OCC based on both neural networks [12–14] and support vector machines (SVMs) [15]. Neural autoencoders, for example, learn to optimally reconstruct acceptable plans by backpropagating reconstruction error through a given network structure of neural units [16]. Once the weights are tuned to reconstruct training plans, testing plans' reconstruction errors are used for classification. Unacceptable plans are expected to exhibit higher reconstruction error than acceptable plans, because the network has not been trained to reconstruct them.

We compare OART to other unsupervised feature extractors applicable to a one-class setting: principal component analysis (PCA) [17], independent component analysis (ICA) [18], and autoencoders (AEs) [19]. One-class classifiers in the literature are usually based on neural networks (restricted Boltzmann machines [20] and AE classifiers [21]), or one-class SVMs (OSVMs) [15,22]. Restricted Boltzmann machines require deep structures and very large training datasets [23], while RT datasets, including ours, generally have small sample sizes. We therefore implement AE and OSVM as classifiers after feature selection with OART.

OSVM algorithms identify anomalies by defining the smallest hyperspace that encloses most of the acceptable plans and classifying any outliers as anomalies. Depending on the approach, this hyperspace can be a sphere [22] or a hyperplane [15], with both methods showing comparable performance [11]. However, OSVM struggles in high-dimensional feature spaces [24], such as those encountered in RT treatment datasets, where the complexity of anatomical, dosimetric, and machine deliverability features increases the risk of overfitting and inaccurate margins. Unlike OSVM, our OART method leverages adaptive resonance theory to dynamically refine feature selection, mitigating overfitting while preserving decision boundaries that align with clinical QA needs. By focusing on stable feature extraction, OART provides a more interpretable and robust approach to anomaly detection in RT planning.

Deep One-Class Classification [25], while being widely used and effective for one-class classification tasks, requires data from both classes during training. In contrast, our algorithm leverages the power of adaptive resonance theory and requires only samples from a single class for training. This characteristic allows our model to excel in scenarios where data from the other class might be scarce or unavailable.

HRN [26], on the other hand, offers a one-class learning approach and can be trained with a single class. However, it employs a multilayer perceptron with a specific cost function, which can make the model less interpretable. In contrast, our model maintains transparency and interpretability, allowing for easier understanding of the decision-making process.

ART is a simplistic mathematical model of feature learning in the biological brain. Since its introduction in 1987 [27], many different variations have been proposed to improve the

existing methodology, or to provide context-specific solutions. ART-based solutions have been used in different machine learning tasks, namely unsupervised, supervised and reinforcement learning [28]. PART [29] is a particular extension of ART, that was developed as a projective version of ART for pattern recognition in high-dimensional spaces. PART-TSP [30,31] was developed later as a supervised version of the projective ART.

The high-dimensionality (with mixed feature types), and class imbalance are the two main challenges in out RT QA problem. An ART-based model solves both problems simultaneously in an interpretable fashion: 1) The adaptive learning process removes reliance on backpropagation and allows for learning from only acceptable samples. 2) ART-based systems provide scalable, fast, and reliable machine learning solutions for mixed-domain input [32] which high dimensionality [33], which is the case for a comprehensive RT QA.

In OART, learning happens as a result of an adaptive resonance procedure that focuses on stable similarities between acceptable samples. The primary contribution of this work is one-class feature extraction that selectively computes stable long-term values and quantifies deviations from them to measure anomaly. One-class learning can happen without any samples required from the unacceptable class, thanks to the adaptive learning that does not require backpropagation. The extracted features may be combined with any OCC, or be used as a standalone OCC with interpretable classification scores.

While ART has been widely studied and applied in various domains, our contribution extends beyond the mere application to radiotherapy treatment planning. OART is the only one-class supervised version of the ART algorithm, specifically designed for anomaly detection. By leveraging the one-class classification framework, OART learns solely from acceptable plans, allowing for interpretable and explainable quality assurance. This adaptation introduces a novel perspective in the field of RT plan classification, contributing to both the advancement of ART and the explainable enhancement of patient safety in radiotherapy treatments.

Since clinical norms and standards may change over time, and the QA system is expected to adapt, we use a variation of adaptive resonance theory (ART) [34] as the base learner in our methodology. Furthermore, such networks learn adaptively from matches, rather than backpropagating mismatches; and are therefore a natural match for our one-class problem statement. Our neural network assigns two weights to each feature: a binary switch to indicate the presence of that feature in the memory and a real-valued weight to store long-term memory. It adaptively learns the values of these two sets of weights according to a resonance procedure that requires samples from only one class. By keeping the features unmixed in the feature extraction phase, OART provides interpretable results, that can be particularly valuable in RT QA, where the planners can use feature-specific feedback for plan quality improvement.

We used two clinical datasets of breast and prostate treatment plans to validate our model. We show that for both datasets, OART outperforms other common unsupervised feature extraction methods in improving the performance of OSVM and AE as one-class classifiers. We also show that OART, as a classifier by itself, provides unique feature-wise interpretability, while achieving only slightly lower AUC than the best performer (OART+AE).

While previous research in binary classification and one-class approaches has shown promise in RT QA, these methods either struggle with novel anomalies or suffer from poor performance in high-dimensional, small-sample datasets. Our method, OART, addresses these limitations by integrating adaptive resonance theory with feature selection, enabling robust and interpretable anomaly detection in RT planning.

## 2 Data

We use two clinically curated radiotherapy datasets of breast and prostate cancer patients. They are two of the most common cancers with 12.5% and 12.2% respectively [35], and thus not only are an impactful group to study but have a lot of data. The breast dataset has 160 treatments (142 acceptable and 18 unacceptable) each with 1,727 numeric features, and the prostate dataset has 434 treatments (421 acceptable and 13 unacceptable) each with 2,697 numeric features, owing to the larger number of OARs in prostate cancer (Table 1).

### 2.1 Plan features

Each plan has three categories of features: ROI features, dose features, and machine features. Table 1 summarizes feature details and feature counts. The ROI features capture the anatomical shape and intensity of the OARs and the tumour(s). If a particular ROI is present in only a subset of the plans, the corresponding ROI features are set to zero for the plans in which that ROI is not present. Breast will include lung, heart, and breast at minimum, and prostate will include prostate, left femur, right femur, bladder, and rectum. The number of ROI and dose features are higher in the prostate dataset, as typically more organs are involved in a prostate case than in a breast case. But since both breast and prostate plans are designed for the same technique (volumetric modulated arc therapy), the number of machine features are similar.

The dosimetric features capture how much beam radiation each ROI receives. Dose volume histograms [37] measure what proportion of the OARs and the tumour(s) receive what proportion of the prescribed dose. Ideally, most of the tumour tissues receive at least a certain amount of radiation, while as little of the healthy organs as possible receive more than the prescribed dose.

**Table 1. Plan features and feature counts.**

| Category (feat. count[1]) | Subcategory | Details |
|---|---|---|
| ROI (1550, 2453) | Shape | centroid |
| | | variation along PCs[2] |
| | | volume |
| | | SDF[3] histogram |
| | | maximum thickness |
| | Intensity | mean |
| | | mode |
| | | standard deviation |
| | | minimum and maximum |
| | | histogram |
| | Joint | shape and dose features |
| Dose (13, 80) | DVH | mean |
| | | mode |
| | | standard deviation |
| | | minimum and maximum |
| Machine (164, 164) | Beam | angle, distance, area |
| | Fraction | number of beams |
| | Plan | number of fractions |
| | | Complexity [4] |

1 Count of features in breast, and prostate datasets, respectively 2 Principal components 3 Signed distance function has positive values inside the OAR and negative values outside [36]. 4 Modulation complexity score [5]

The machine features capture the radiation variables from the source's point of view. Each plan is divided into several fractions, each of which will be delivered in a separate session. Each fraction, then, includes a set of beams whose intensities, distances to ROI surfaces, and effective areas are calculated. An overall complexity score [5] is also included to reflect deliverability complexities. Plans with high complexity are undesirable.

Since the number of beams and control points vary per plan, beam and control point features are variable in length, but all records should contain the same number of features for classification. We therefore used a bag-of-words model combined with dictionary learning [38] to map these variable number of beam and control point related features to a constant length feature space based on their membership to unsupervised clusters of data points in full dimensionality. The bag-of-words model can be split into a training phase, where the system learns a dictionary of basis functions, and an encoding phase, where the dictionary is used to extract features from new inputs.

## 2.2 Plan samples

Let $T$ be the set of all available plans that are all labelled as acceptable (-1) or unacceptable (+1) by clinical experts.

Unlike binary classification, during training for one-class classification, all unacceptable plans ($T^+$) are left out of training dataset and are only used in the testing dataset. The acceptable plans ($T^-$) are randomly split (80%-20%) into $T^-_{\text{train}}$ and $T^-_{\text{test}}$. Therefore, the training is performed on only the acceptable plans, while testing is performed on a mix of acceptable and unacceptable plans.

## 3 Methods

### 3.1 Data acquisition and ethical considerations

The datasets were obtained over three years from Princess Margaret Cancer Centre (Toronto, Canada) under an institutional ethics-approved research protocol. We obtained Research Ethics Board (REB) approval from the University of Toronto. The protocol reference number is 14-8165. Since we used retrospective study of medical records or archived samples and the data was de-identified (anonymized), no patient identifying information is available. The need for consent was waived.

### 3.2 OART structure and training

The goal of our training approach is to adaptively learn from samples that resonate with long-term memory. We utilize an ART-based neural network structure, called OART, to form a stable and adaptable long-term memory of acceptable plans. By measuring the deviation of a new plan $\mathbf{x}$ from this memory, we can predict its acceptance or rejection.

OART is a neural network consisting of two sets of weights: a vector of boolean switches $\mathbf{f} \in \{0,1\}^m$ and a vector of real-valued long-term memories $\mathbf{z} \in [0,1]^m$, where $m$ is the number of original input features. Each input dimension $i = 1, \dots, m$ has a switch $\mathbf{f}[i]$ that acts as a gate for the sensory input to reach the long-term memory for comparison. This mechanism allows the network to focus its attention on a subset of features with stable memories and discard the rest. The real-valued $\mathbf{z}[i]$ represents the long-term memory for feature $i$ and is adaptively updated during resonance learning.

To mitigate the order dependence in vanilla adaptive learning, we adopt an unsupervised implementation of ART with distributed dual vigilance [39]. For our one-class problem, we repeat the training for $E$ rounds of randomly-ordered training plans. At each round, the

training starts with a shuffled full epoch of training plans and continues until the number of surviving features stabilizes (i.e., does not increase by more than 0.1%).

At the beginning of each training round, the memory values are set to the first training plan, and all feature switches are initially active:

$$\mathbf{z}[i] = T^{-}_{\text{train}}[1, i] \quad i = 1, \dots, m$$
$$\mathbf{f}[i] = 1 \quad i = 1, \dots, m$$

Here, $T^{-}_{\text{train}}$ is the training matrix where the first index represents the sample, and the second index represents the feature.

During each training step, learning occurs through a resonance check between each feature of the old memory and that of the new plan, followed by an adaptive update of the memory towards resonant features of the new plan. Only features that match (i.e., are within a distance of $\sigma$) between the new plan and the old memory remain active, while the rest are deactivated ($\mathbf{f}[i] = 0$, $i \notin \{i : |\mathbf{x}[i] - \mathbf{z}[i]| \le \sigma\}$) for the remainder of that training round. Resonant features undergo an adaptation procedure where the long-term memory is shifted towards the new plan. Another plan is then presented as input in the next training step. Fig 1 shows the structure of OART.

For a given distance parameter $\sigma$, let $\mathbf{z}_\sigma$ and $\mathbf{f}_\sigma$ denote the weights. Then the learning equations are as follows:

$$\mathbf{z}^{\text{new}}_\sigma = (1 - \alpha)\mathbf{z}^{\text{old}}_\sigma + \alpha\mathbf{x} \tag{1}$$

$$\mathbf{f}^{\text{new}}_\sigma = \mathbf{f}^{\text{old}}_\sigma \odot \mathbf{h}_\sigma(\mathbf{x}, \mathbf{z}^{\text{old}})$$

where $\alpha \in (0, 1)$ is the learning rate, $\odot$ denotes element-wise multiplication, and

$$\mathbf{h}_\sigma(\mathbf{x}, \mathbf{z}^{\text{old}})[i] = \begin{cases} 1 & \text{if } |\mathbf{x}[i] - \mathbf{z}^{\text{old}}[i]| \le \sigma \\ 0 & \text{otherwise} \end{cases} \quad i = 1, \dots, m$$

performs the similarity check.

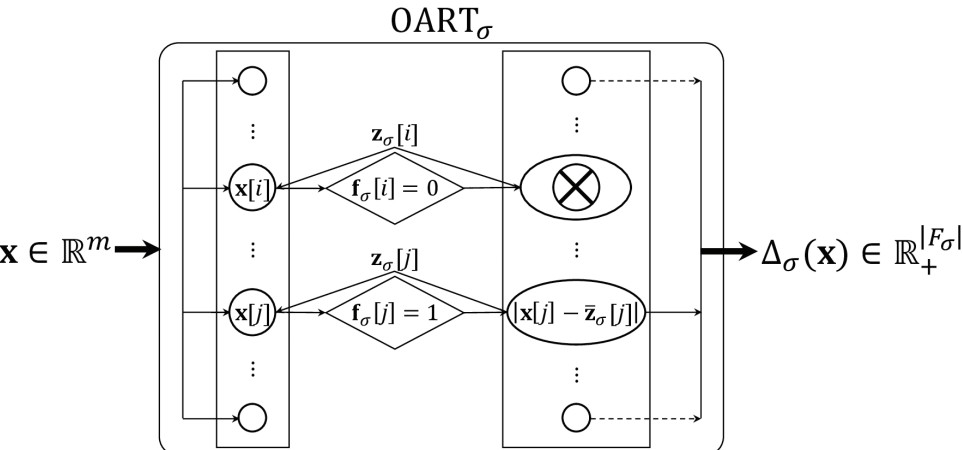

Fig 1. Structure of OART$_\sigma$. Bold arrow indicates vectors and regular arrow indicates scalars.

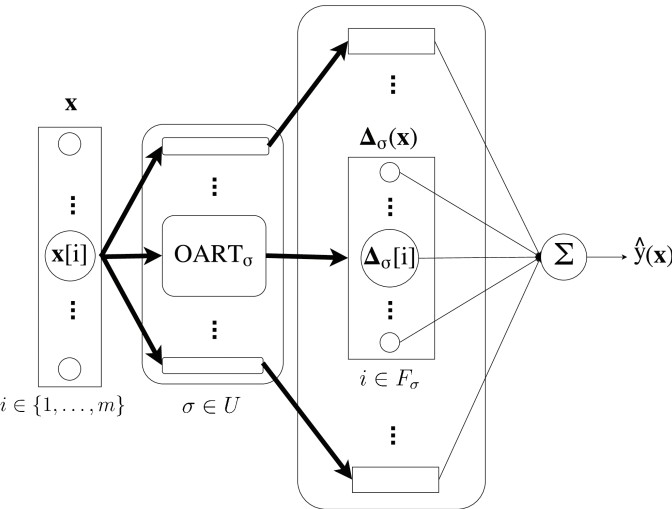

**Fig 2. OART as standalone classifier OART** + $\Sigma$. Bold arrow indicates vectors and regular arrow indicates scalars.

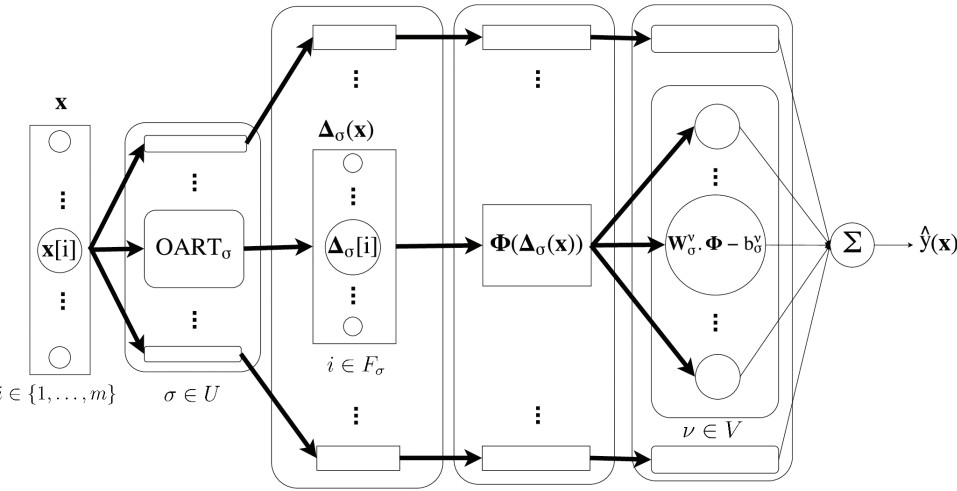

**Fig 3. OART as feature extractor: OART + OSVM.** Bold arrow indicates vectors and regular arrow indicates scalars.

### 3.3 Feature extraction and classification

At the end of each training round $e = 1, \dots, E$, the terminal values of the weights provide the survived features and their corresponding values, denoted as $\mathbf{f}_\sigma^e$ and $\mathbf{z}_\sigma^e$, respectively. Features that survive in more than a desired proportion $\gamma$ of training rounds are considered stable features and their memory values are extracted as the average of the terminal training values across all rounds (Figs 5 and 6). We define $F_\sigma$ as the set of stable features:

$$F_\sigma = \left\{ i : \sum_{e=1}^{E} \mathbf{f}_\sigma^e[i] \geq \gamma E, \; i = 1, \dots, m \right\} \tag{2}$$

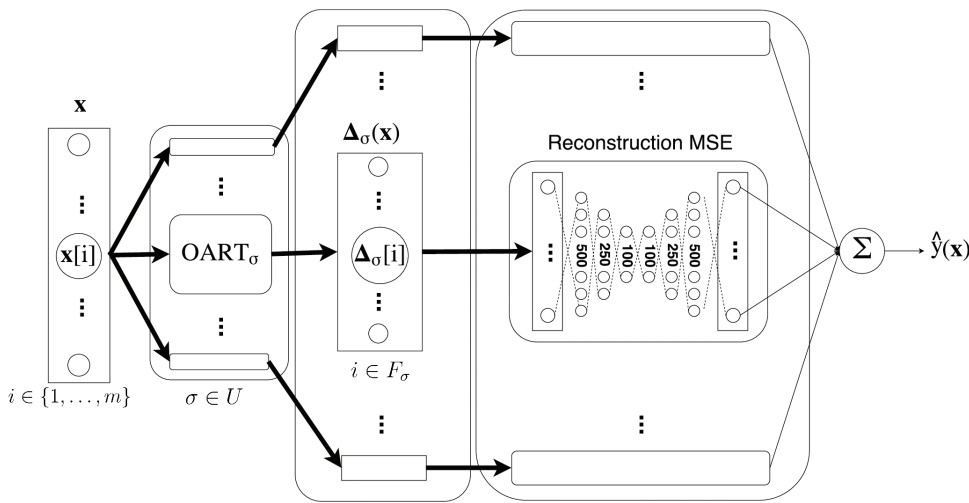

**Fig 4. OART as feature extractor: OART + AE.** Bold arrow indicates vectors and regular arrow indicates scalars.

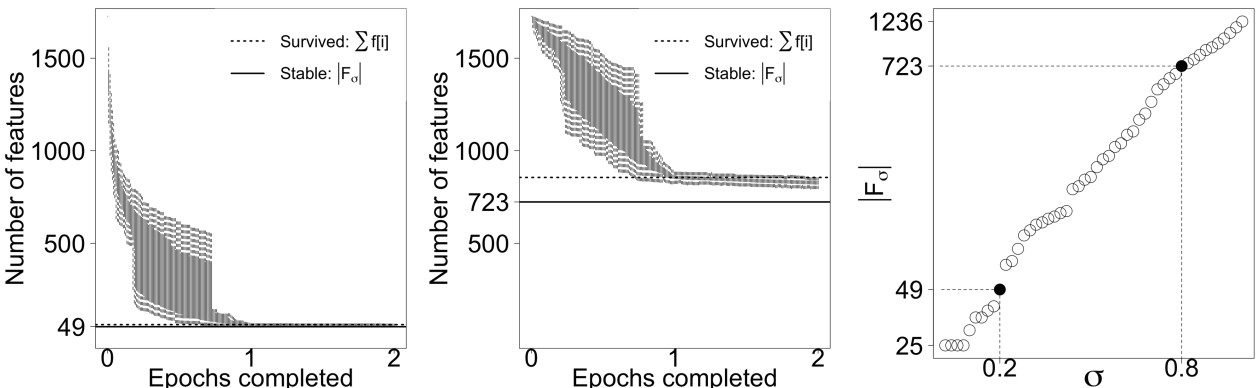

**Fig 5. Feature survival for breast.** Left, middle: The number of survived features are for $\sigma = 0.2$ (left) and $\sigma = 0.8$ (middle) over different runs. Dashed lines show number of features that survived a particular run as an example. Solid lines show the count of features who survived most of the runs, and therefore are labelled stable. Right: The number of stable features at each value of $\sigma \in \{0.01, 0.03, \dots, 0.99\}$, with $\sigma = 0.19$ and $\sigma = 0.79$ denoted 0.2 and 0.8 on the $x$-axis.

$$\bar{\mathbf{z}}_\sigma[i] = \frac{1}{E}\sum_{e=1}^{E}\mathbf{z}_\sigma^e[i] \qquad i = 1, \dots, m$$

$$\bar{\mathbf{f}}_\sigma[i] = \begin{cases} 1 & i \in F_\sigma \\ 0 & \text{otherwise} \end{cases} \qquad i = 1, \dots, m$$

To measure the deviation of a plan $\mathbf{x}$ from the memory values $\bar{\mathbf{z}}_\sigma$ of the features in $F_\sigma$, we define $\Delta_\sigma(\mathbf{x})$ as follows:

$$\Delta_\sigma(\mathbf{x})[i] = |\mathbf{x}[i] - \bar{\mathbf{z}}_\sigma[i]| \quad i \in F_\sigma \tag{3}$$

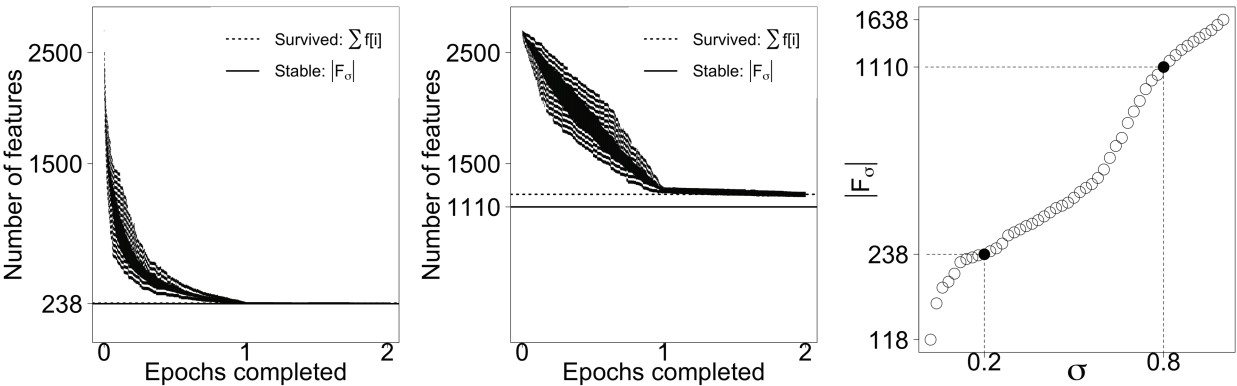

**Fig 6. Feature survival for prostate.** Left, middle: The number of survived features are for $\sigma = 0.2$ (left) and $\sigma = 0.8$ (middle) over different runs. Dashed lines show number of features that survived a particular run as an example. Solid lines show the count of features who survived most of the runs, and therefore are labelled stable. Right: The number of stable features at each value of $\sigma \in \{0.01, 0.03, \ldots, 0.99\}$, with $\sigma = 0.19$ and $\sigma = 0.79$ denoted 0.2 and 0.8 on the $x$-axis.

The classification is then performed on the deviations $\Delta_\sigma(\mathbf{x})$, which reside in $\mathbb{R}_+^{|F_\sigma|}$. We boost the classification results by adding classification scores over a set of $\sigma$ values, where smaller values represent tighter similarity conditions, resulting in a smaller number of features surviving in stable memory.

It is important to note that there are many variants of autoencoders, each with its own unique characteristics and use cases. For example, convolutional autoencoders [40] are designed for image data, while recurrent autoencoders [41] are designed for sequential data. Sparse autoencoders [42] are designed to learn sparse representations of the input data, while contractive autoencoders [43] are designed to be more robust to small perturbations in the input data. Laplacian autoencoders [44] are designed to learn a low-dimensional representation of the input data that preserves the local structure of the data. However, we only used a vanilla as a representative, because our main claimed advantage of OART is explainability, which is absent in all AE variants as a result of their multilayer perceptron structure.

In the case of using an Autoencoder for one-class classification, we follow a two-step process. Firstly, we employ the AE as a feature extractor by cutting the network at the middle layer, which serves as the bottleneck layer capturing the deep features of the input data. Once the deep features are extracted, we utilize the AE as a classifier. To make one-class classification decisions, we use the reconstruction mean squared error (MSE) as the classification score. Outlier instances are expected to exhibit higher MSE during the reconstruction process.

To determine the classification threshold, we conduct testing on a set of labeled data. By varying the threshold from the minimum to the maximum MSE observed during testing, we can explore different trade-offs between false positives and false negatives.

The OSVM was implemented in the libsvm package with Gaussian kernel (we excluded linear, polynomial, and sigmoid kernels, as none provided AUC larger than 0.5). The algorithm looks for the smallest hyperspace in the kernel space that contains most of training plans. Let $\Phi$ be a feature map $\mathbb{R}_+^{|F_\sigma|} \to \mathcal{F}$, i.e., a map from $\Delta_\sigma$ space into an inner product space $\mathcal{F}$ such that the inner product can be computed by evaluating some simple kernel [45]. Then, the separating hyperspace is found by solving the following optimization problem:

$$\underset{\mathbf{W}\in\mathcal{F},\, \mathbf{s}\in\mathbb{R}_+^n,\, b\in\mathbb{R}}{\text{minimize}} \quad \frac{1}{2}||\mathbf{W}||^2 + \frac{1}{\nu \times n}\sum_{\eta=1}^{n}(\mathbf{s}[\eta] - b)$$

$$\text{subject to} \quad \mathbf{W} \cdot \Phi\big(\Delta_\sigma(\mathbf{x})\big) \geq b - \mathbf{s}[\eta]$$

where $b$ is the bias, $\mathbf{s}$ is the vector of the slack variables, and $\nu$ is a parameter in (0,1] that trades off between the two objective terms: hyperspace and the proportion of acceptable plans to encompass. Note that a positive dot product $\mathbf{W} \cdot \Phi(\Delta(\mathbf{x}))$ indicates that $\mathbf{x}$ is within acceptable hyperspace and therefore its class label is acceptable, and vice versa.

Let $\mathbf{W}_\sigma^\nu$ and $b_\sigma^\nu$ denote the optimal values of $\mathbf{W}$ and $b$ for distance parameter $\sigma$ and model complexity $\nu$. The decision boundary is then

$$S_\sigma^\nu(\mathbf{x}) = \mathbf{W}_\sigma^\nu \cdot \Phi\big(\Delta_\sigma(\mathbf{x})\big) - b_\sigma^\nu$$

which is positive for most training acceptable plans. Unacceptable plans should have $S_\sigma^\nu(\mathbf{x}) < 0$. The magnitude indicates how far from the boundary the plan is. Different values $\nu \in V = \{0.01, 0.02, \dots, 0.99\}$ are considered. Classification scores from distance parameters $\sigma \in U$ and complexities $\nu \in V$ are aggregated by summation into a single score:

$$\overline{S}(\mathbf{x}) = \sum_{\nu \in V}\sum_{\sigma \in U} S_\sigma^\nu(\mathbf{x})$$

## 4 Results

We first present the classification results obtained from OSVM and AE classifiers with and without applying feature extractors. Then, we present the clinical interpretations of classification on OART's features.

### 4.1 Classification results

Our MATLAB implementation of OART is available on github, and the link is here. We ran OART on a 3.5 GHz Intel Core i7 processor with 16GB RAM.

We utilized min-max scaling to normalize the features. This normalization technique was chosen to ensure that all features have a consistent scale, thus preventing any particular feature from dominating the classification process.

Although the breast dataset has fewer features, it has so many fewer samples that its record-to-feature ratio is still lower than that of the prostate dataset. There are 119 training samples for breast with 1,727 features (a record-to-feature ratio of 6.89%), and 345 training samples for prostate with 2,697 features (record-to-feature ratio of 12.79%). In Sect 5, we discuss the potential repercussions of the breast dataset's especially sparse training space.

The values of parameters $\alpha$ and $\gamma$ cannot be optimized end-to-end because there is no binary classification error metric. The value of $\alpha$, the adaptive learning rate in Eq 1, is set to a small number ($\alpha = 0.1$) to only slightly shift the long-term memory of OART towards the new resonant plans. The value of $\gamma$, the survival stability rate in Eq 2 is set to a large number ($\gamma = 0.9$) to keep the features that survived almost all training shuffles (the difference is highlighted by solid line and the dashed line in Fig 5 for breast; and Fig 6 for prostate).

Training iterations stop when the marginal change is less than 0.1%. In a single training round $e = 1, \dots, E$, the size of survived dimensions $\sum_{i=1}^{m} \mathbf{f}[i]$ decreases by less than 0.1% before the end of the second epoch (Figs 5 and 6), so each training round $e$ includes two shuffled epochs of training plans. For breast and prostate, the size of the stable features $|F_\sigma|$ changes by

less than 0.1% after 12 and 17 training rounds, respectively, so we set $E = 20$ as a conservative number of epochs to reach convergence.

Our approach involves a two-pronged strategy for applying the $\sigma$ values. Initially, the data features are linearly normalized into the range [0,1]. Given that the distance parameter dictates the degree of similarity necessary for resonance learning, we compare two cases: learning from a full range between 0 and 1, in increments of 0.02 ($U = 0.01, 0.03, \ldots, 0.99$) and learning only from the more sensitive (smaller) $\sigma$ values ($\tilde{U} = 0.01, 0.03, \ldots, 0.49$). Each $\sigma$ value, regardless of being larger or smaller, has the potential to offer valuable insights. Without comprehensive binary validation, it would be presumptuous to deem any $\sigma$ values as 'useless', hence our inclusive approach.

Once OART features are extracted for various $\sigma \in U$, we have two paths for classification. Firstly, the raw sum of deviations can be treated as a classification score, providing an intuitive yet effective classification criterion - higher total deviations suggest a higher likelihood of an entity being an outlier. This method ensures interpretability of the classification score. Alternatively, these raw features can be forwarded to classifiers like OSVM and the AE. While this could offer a slight gain in accuracy, it comes at the expense of explainability. Regardless of the path chosen, the scores from different $\sigma$ values are aggregated, hence the use of summation.

For lower $\sigma$ values, most of stable features are machine features and for larger $\sigma$ values, ROI and dose features are also included in the stable memory (Figs 5 and 6). Machine features of the acceptable plans are therefore expected not to deviate much from memory values. Testing plans that exhibit large deviations on machine features are likely to be classified as unacceptable. Larger $\sigma$ means more tolerance to deviation from memory, larger number of survived dimensions in each training round, and therefore more stable features. Both breast and prostate plans exhibit similar performance, so only prostate plots are shown.

It is essential to clarify the data shuffling method employed in our study. Initially, the entire set of training samples is shuffled, forming the first epoch. Subsequently, another epoch is shuffled and appended, resulting in a sequence of twice the size, in which each data point appears once in the first half and once in the second. This procedure ensures a fair distribution and exposure of data points throughout the training process. The observed plateau in Figs 5 and 6 is indicative of the model's learning behavior that is explained below. Essentially, the model size stabilizes after the first epoch, reflecting that the first epoch is sufficient to extract the significant patterns from the data. The consistency of this behavior across all training rounds suggests that the algorithm's learning is not significantly boosted by subsequent passes over the data, hence the stabilization of model size during the second epoch, and no need for further epochs.

OART performance is compared to PCA, ICA, and AEs. Since the number of plans is smaller than the number of original features, PCA and ICA provide as many components as the samples. as the samples. All of these components are included in the classification. We determined the size and structure of the autoencoder based on the reconstruction error. We considered three structures: shallow, deep, and bottleneck using 50, 100, 200, 400, 800, and 1600 units with $\tan^{-1}$ activation function and dropout regularization. The reconstruction error for all structures and unit numbers is 0.026 with $< 0.1\%$ variation between structures. We selected the structure and unit size with the lowest reconstruction MSE (bottleneck with 1600 units).

The features from each feature extraction method are then inputted into OSVM and AE for one-class classification (e.g., PCA+OSVM and OART+AE). Since OART's features indicate deviation from stable memory values, we also performed classification based on the sum of OART's feature values (OART+$\Sigma$). Tables 2 and 3 summarize the performance metrics.

**Table 2. Breast: 95% confidence intervals for performance metrics. Highlight indicates best performance in each dataset.**

| | Features | Classifier | $Spec_{100}$ | $Spec_{80}$ | $Sens_{100}$ | $Sens_{80}$ | AUC |
|---|---|---|---|---|---|---|---|
| Binary | original | LR | 0 | 0 | $0.19_{\pm 0.08}$ | $0.55_{\pm 0.03}$ | $0.75_{\pm 0.02}$ |
| | | RF | $0.16_{\pm 0.03}$ | $0.39_{\pm 0.04}$ | $0.35_{\pm 0.04}$ | $0.42_{\pm 0.04}$ | $0.65_{\pm 0.03}$ |
| | | SVM | 0 | $0.06_{\pm 0.06}$ | $0.21_{\pm 0.08}$ | $0.60_{\pm 0.03}$ | $0.78_{\pm 0.02}$ |
| | | NB | 0 | 0 | $0.03_{\pm 0.04}$ | $0.42_{\pm 0.04}$ | $0.64_{\pm 0.02}$ |
| | | NN | 0 | 0 | $0.14_{\pm 0.03}$ | $0.53_{\pm 0.03}$ | $0.72_{\pm 0.02}$ |
| | LASSO | LR | 0 | 0 | 0 | $0.21_{\pm 0.03}$ | $0.53_{\pm 0.08}$ |
| | PCA | LR | 0 | 0 | $0.15_{\pm 0.06}$ | $0.57_{\pm 0.03}$ | $0.73_{\pm 0.02}$ |
| | | RF | $0.12_{\pm 0.07}$ | $0.20_{\pm 0.05}$ | $0.34_{\pm 0.03}$ | $0.45_{\pm 0.02}$ | $0.67_{\pm 0.02}$ |
| | | SVM | 0 | 0 | $0.18_{\pm 0.04}$ | $0.53_{\pm 0.04}$ | $0.73_{\pm 0.03}$ |
| | | NB | $0.05_{\pm 0.06}$ | $0.07_{\pm 0.02}$ | $0.10_{\pm 0.02}$ | $0.42_{\pm 0.02}$ | $0.66_{\pm 0.02}$ |
| | | NN | 0 | 0 | 0 | $0.32_{\pm 0.04}$ | $0.68_{\pm 0.02}$ |
| | ICA | LR | 0 | 0 | $0.15_{\pm 0.05}$ | $0.60_{\pm 0.03}$ | $0.75_{\pm 0.02}$ |
| | | RF | $0.10_{\pm 0.03}$ | $0.27_{\pm 0.04}$ | $0.30_{\pm 0.03}$ | $0.45_{\pm 0.02}$ | $0.65_{\pm 0.03}$ |
| | | SVM | 0 | 0 | $0.19_{\pm 0.04}$ | $0.50_{\pm 0.02}$ | $0.76_{\pm 0.03}$ |
| | | NB | 0 | 0 | $0.10_{\pm 0.07}$ | $0.49_{\pm 0.04}$ | $0.73_{\pm 0.02}$ |
| | | NN | 0 | 0 | $0.05_{\pm 0.05}$ | $0.30_{\pm 0.02}$ | $0.63_{\pm 0.02}$ |
| | AE | LR | 0 | $0.11_{\pm 0.03}$ | 0 | $0.38_{\pm 0.03}$ | $0.63_{\pm 0.02}$ |
| | | RF | $0.08_{\pm 0.04}$ | $0.21_{\pm 0.03}$ | $0.23_{\pm 0.03}$ | $0.36_{\pm 0.02}$ | $0.65_{\pm 0.03}$ |
| | | SVM | $0.07_{\pm 0.05}$ | $0.22_{\pm 0.03}$ | $0.31_{\pm 0.02}$ | $0.44_{\pm 0.03}$ | $0.78_{\pm 0.02}$ |
| | | NB | 0 | 0 | $0.07_{\pm 0.03}$ | $0.30_{\pm 0.03}$ | $0.73_{\pm 0.02}$ |
| | | NN | 0 | $0.14_{\pm 0.03}$ | $0.23_{\pm 0.02}$ | $0.39_{\pm 0.02}$ | $0.75_{\pm 0.02}$ |
| OCC | original | AE | $0.07_{\pm 0.03}$ | $0.53_{\pm 0.06}$ | $0.05_{\pm 0.03}$ | $0.66_{\pm 0.04}$ | $0.73_{\pm 0.02}$ |
| | | OSVM | $0.09_{\pm 0.02}$ | $0.33_{\pm 0.03}$ | $0.08_{\pm 0.03}$ | $0.48_{\pm 0.03}$ | $0.65_{\pm 0.02}$ |
| | PCA | AE | $0.07_{\pm 0.02}$ | $0.23_{\pm 0.04}$ | $0.24_{\pm 0.04}$ | $0.36_{\pm 0.03}$ | $0.59_{\pm 0.02}$ |
| | | OSVM | $0.03_{\pm 0.02}$ | $0.41_{\pm 0.05}$ | $0.05_{\pm 0.03}$ | $0.49_{\pm 0.04}$ | $0.66_{\pm 0.02}$ |
| | ICA | AE | $0.11_{\pm 0.02}$ | $0.38_{\pm 0.04}$ | $0.23_{\pm 0.03}$ | $0.47_{\pm 0.03}$ | $0.69_{\pm 0.02}$ |
| | | OSVM | $0.00_{\pm 0.01}$ | $0.20_{\pm 0.04}$ | $0.22_{\pm 0.02}$ | $0.42_{\pm 0.03}$ | $0.60_{\pm 0.02}$ |
| | AE | OSVM | $0.20_{\pm 0.03}$ | $0.51_{\pm 0.03}$ | $0.06_{\pm 0.03}$ | $0.52_{\pm 0.03}$ | $0.72_{\pm 0.02}$ |
| | $OART^U$ | $\Sigma$ | $0.04_{\pm 0.02}$ | $0.35_{\pm 0.05}$ | $0.22_{\pm 0.05}$ | $0.63_{\pm 0.03}$ | $0.70_{\pm 0.02}$ |
| | | AE | $0.26_{\pm 0.05}$ | $0.73_{\pm 0.04}$ | $0.10_{\pm 0.05}$ | $0.78_{\pm 0.03}$ | $0.81_{\pm 0.01}$ |
| | | OSVM | $0.11_{\pm 0.04}$ | $0.57_{\pm 0.05}$ | $0.08_{\pm 0.03}$ | $0.60_{\pm 0.04}$ | $0.74_{\pm 0.02}$ |
| | $OART^{\bar{U}}$ | $\Sigma$ | $0.24_{\pm 0.02}$ | $0.58_{\pm 0.05}$ | $0.24_{\pm 0.03}$ | $0.69_{\pm 0.03}$ | $0.78_{\pm 0.02}$ |
| | | AE | $0.34_{\pm 0.02}$ | $0.67_{\pm 0.05}$ | $0.11_{\pm 0.05}$ | $0.75_{\pm 0.03}$ | $0.80_{\pm 0.01}$ |
| | | OSVM | $0.35_{\pm 0.03}$ | $0.67_{\pm 0.04}$ | $0.05_{\pm 0.02}$ | $0.66_{\pm 0.04}$ | $0.78_{\pm 0.01}$ |

In binary classification, testing metrics are derived through leave-one-out cross validation, where training occurs on all data from both classes except for the sample currently being tested. Consequently, these metrics are not directly comparable to those of One-Class Classification (OCC), where only 80% of the negative samples are used. Nevertheless, the one-class scenario involves utilizing less training data for each test sample, making it technically more challenging.

The performance comparison in our study includes five metrics that assess the machine learning model's performance at extreme misclassification costs (100%) and at the average human vigilance level in industrial Quality Assurance (QA) settings (80% [46]). Specifically, we consider two metrics: Sens80 and Spec80.

$Sens_{80}$ represents the maximum sensitivity of the model when it maintains at least 80% specificity. It indicates the highest percentage of unacceptable plans that can be correctly detected while ensuring that at least 80% of acceptable plans are identified accurately. This metric allows us to evaluate the model's performance in scenarios where maintaining high specificity is crucial.

Similarly, Spec80 represents the maximum specificity of the model when it maintains at least 80% sensitivity. It measures the percentage of acceptable plans that can be accepted

**Table 3. Prostate: 95% confidence intervals for performance metrics. Highlight indicates best performance in each dataset.**

| | Features | Classifier | $Spec_{100}$ | $Spec_{80}$ | $Sens_{100}$ | $Sens_{80}$ | AUC |
|---|---|---|---|---|---|---|---|
| Binary | original | LR | 0 | $0.03_{\pm 0.05}$ | $0.06_{\pm 0.04}$ | $0.53_{\pm 0.04}$ | $0.70_{\pm 0.02}$ |
| | | RF | $0.25_{\pm 0.03}$ | $0.51_{\pm 0.07}$ | $0.35_{\pm 0.06}$ | $0.70_{\pm 0.04}$ | $0.79_{\pm 0.02}$ |
| | | SVM | 0 | 0 | $0.17_{\pm 0.06}$ | $0.44_{\pm 0.03}$ | $0.71_{\pm 0.02}$ |
| | | NB | 0 | 0 | 0 | $0.29_{\pm 0.03}$ | $0.61_{\pm 0.02}$ |
| | | NN | $0.14_{\pm 0.03}$ | $0.28_{\pm 0.03}$ | $0.34_{\pm 0.03}$ | $0.69_{\pm 0.02}$ | $0.81_{\pm 0.03}$ |
| | LASSO | LR | 0 | 0 | 0 | $0.32_{\pm 0.06}$ | $0.63_{\pm 0.07}$ |
| | PCA | LR | 0 | 0 | 0 | $0.45_{\pm 0.03}$ | $0.69_{\pm 0.03}$ |
| | | RF | $0.06_{\pm 0.03}$ | $0.11_{\pm 0.03}$ | 0 | $0.23_{\pm 0.02}$ | $0.65_{\pm 0.03}$ |
| | | SVM | 0 | 0 | $0.05_{\pm 0.03}$ | $0.29_{\pm 0.02}$ | $0.67_{\pm 0.03}$ |
| | | NB | 0 | 0 | 0 | $0.14_{\pm 0.02}$ | $0.60_{\pm 0.02}$ |
| | | NN | 0 | 0 | $0.09_{\pm 0.03}$ | $0.29_{\pm 0.02}$ | $0.67_{\pm 0.02}$ |
| | ICA | LR | 0 | 0 | $0.16_{\pm 0.03}$ | $0.47_{\pm 0.02}$ | $0.72_{\pm 0.02}$ |
| | | RF | $0.07_{\pm 0.02}$ | $0.12_{\pm 0.03}$ | $0.13_{\pm 0.02}$ | $0.39_{\pm 0.02}$ | $0.69_{\pm 0.03}$ |
| | | SVM | 0 | 0 | $0.13_{\pm 0.03}$ | $0.39_{\pm 0.02}$ | $0.70_{\pm 0.04}$ |
| | | NB | 0 | $0.05_{\pm 0.03}$ | $0.07_{\pm 0.06}$ | $0.53_{\pm 0.03}$ | $0.73_{\pm 0.02}$ |
| | | NN | 0 | 0 | 0 | $0.35_{\pm 0.02}$ | $0.67_{\pm 0.03}$ |
| | AE | LR | 0 | 0 | $0.23_{\pm 0.04}$ | $0.57_{\pm 0.03}$ | $0.75_{\pm 0.02}$ |
| | | RF | $0.10_{\pm 0.02}$ | $0.16_{\pm 0.03}$ | $0.12_{\pm 0.03}$ | $0.41_{\pm 0.03}$ | $0.67_{\pm 0.03}$ |
| | | SVM | $0.21_{\pm 0.03}$ | $0.48_{\pm 0.03}$ | $0.31_{\pm 0.03}$ | $0.75_{\pm 0.02}$ | $0.80_{\pm 0.03}$ |
| | | NB | 0 | 0 | $0.12_{\pm 0.06}$ | $0.39_{\pm 0.04}$ | $0.73_{\pm 0.02}$ |
| | | NN | $0.24_{\pm 0.02}$ | $0.41_{\pm 0.03}$ | $0.33_{\pm 0.02}$ | $0.60_{\pm 0.02}$ | $0.78_{\pm 0.03}$ |
| OCC | original | AE | $0.26_{\pm 0.03}$ | $0.55_{\pm 0.05}$ | $0.53_{\pm 0.03}$ | $0.69_{\pm 0.03}$ | $0.81_{\pm 0.02}$ |
| | | OSVM | $0.27_{\pm 0.02}$ | $0.48_{\pm 0.04}$ | $0.61_{\pm 0.03}$ | $0.65_{\pm 0.03}$ | $0.78_{\pm 0.02}$ |
| | PCA | AE | $0.02_{\pm 0.01}$ | $0.20_{\pm 0.04}$ | $0.19_{\pm 0.02}$ | $0.39_{\pm 0.03}$ | $0.59_{\pm 0.02}$ |
| | | OSVM | $0.16_{\pm 0.03}$ | $0.40_{\pm 0.04}$ | $0.23_{\pm 0.03}$ | $0.36_{\pm 0.03}$ | $0.65_{\pm 0.02}$ |
| | ICA | AE | $0.01_{\pm 0.01}$ | $0.11_{\pm 0.03}$ | $0.36_{\pm 0.04}$ | $0.49_{\pm 0.03}$ | $0.59_{\pm 0.02}$ |
| | | OSVM | $0.05_{\pm 0.03}$ | $0.26_{\pm 0.04}$ | $0.09_{\pm 0.03}$ | $0.15_{\pm 0.03}$ | $0.53_{\pm 0.02}$ |
| | AE | OSVM | $0.15_{\pm 0.04}$ | $0.46_{\pm 0.04}$ | $0.23_{\pm 0.03}$ | $0.40_{\pm 0.03}$ | $0.68_{\pm 0.02}$ |
| | $OART^{U}$ | Σ | $0.23_{\pm 0.04}$ | $0.63_{\pm 0.06}$ | $0.55_{\pm 0.04}$ | $0.74_{\pm 0.04}$ | $0.81_{\pm 0.03}$ |
| | | AE | $0.80_{\pm 0.01}$ | $0.80_{\pm 0.01}$ | $0.00_{\pm 0.00}$ | $0.75_{\pm 0.05}$ | $0.86_{\pm 0.01}$ |
| | | OSVM | $0.48_{\pm 0.02}$ | $0.67_{\pm 0.04}$ | $0.58_{\pm 0.03}$ | $0.71_{\pm 0.03}$ | $0.81_{\pm 0.01}$ |
| | $OART^{\bar{U}}$ | Σ | $0.32_{\pm 0.05}$ | $0.74_{\pm 0.04}$ | $0.47_{\pm 0.04}$ | $0.76_{\pm 0.03}$ | $0.86_{\pm 0.01}$ |
| | | AE | $0.81_{\pm 0.01}$ | $0.81_{\pm 0.01}$ | $0.00_{\pm 0.00}$ | $0.82_{\pm 0.05}$ | $0.87_{\pm 0.01}$ |
| | | OSVM | $0.29_{\pm 0.06}$ | $0.71_{\pm 0.04}$ | $0.50_{\pm 0.03}$ | $0.71_{\pm 0.03}$ | $0.85_{\pm 0.02}$ |

without human review while minimizing the risk of accepting unacceptable plans. In the context of real-time Quality Assurance (QA), Spec80 is particularly important, as it indicates the level of confidence in accepting plans without manual inspection.

We also consider Sens100 and Spec100, which represent the maximum sensitivity and specificity, respectively, when the model is required to be 100% specific or sensitive. These metrics offer insights into the model's performance under strict conditions where there is no tolerance for accepting misclassified plans.

Additionally, AUC provides an overall comparison by measuring the area under the operating characteristic curve. Testing classification scores are bootstrapped with replacement 100 times and a 95% $t$-distribution confidence interval is calculated for the mean of each of the performance metrics. The ROC curves and performance metrics are shown in Figs 7 and 8.

OART-based methods as feature extractors improved the performance of AE and OSVM classifiers. OART's improvements are the highest among other feature extractors for both breast and prostate datasets. The weak performances of PCA and ICA were expected as both of these methods exploit the variation in the data, while one-class classification is intrinsically based on the similarities of the existing training samples from the one class.

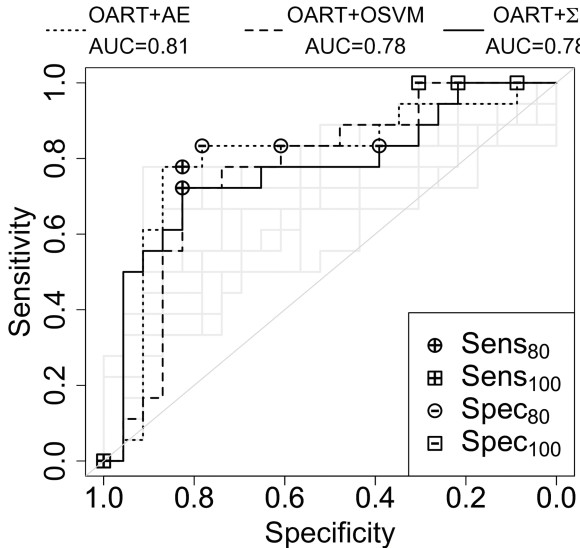

**Fig 7. Receiver operating characteristic curve for the breast dataset.** The highest performing algorithm in black and the rest are grey. The specific OART implemention illustrated is the better of $OART^U$ and $OART^{\bar{U}}$.

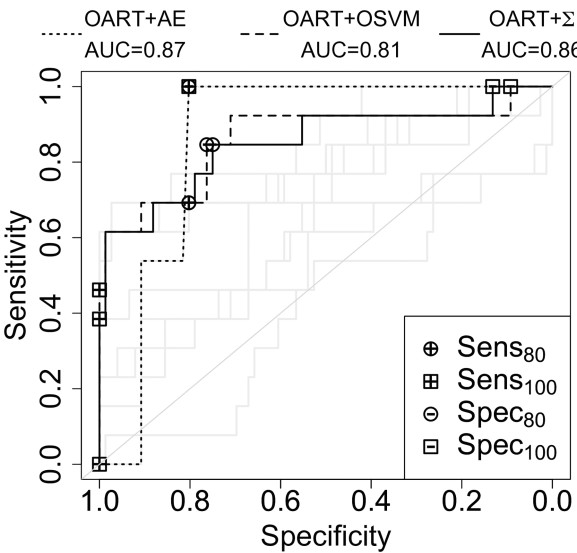

**Fig 8. Receiver operating characteristic curve for the prostate dataset.** The highest performing algorithm in black and the rest are grey. The specific OART implemention illustrated is the better of $OART^U$ and $OART^{\bar{U}}$.

Note that $OART^{\bar{U}} + \Sigma$ achieves higher AUC than $OART^U + \Sigma$ (0.78 compared to 0.70 in breast, and 0.86 compared to 0.81 for prostate) by focusing on the more sensitive half of OART learners. Overall, $OART + \Sigma$ slightly underperforms the top algorithm, $OART + AE$ (0.78 compared to 0.70 in breast, and 0.86 compared to 0.81 for prostate), but it provides feature-wise classification interpretability for clinical purposes, as described in Sect 4.2.

## 4.2 Clinical interpretation

Both autoencoders and support vector machines perform a joint analysis of input features for classification. Their classification scores cannot be directly traced back to the individual input features for interpretation. More specifically, when AE or OSVM classify a plan as unacceptable, they cannot tell which features caused the classifier to classify as such. In a clinical setting, however, transparency and interpretability of classification results are invaluable features for a classifier to have as they can help clinicians initiate targeted improvements to the unacceptable plans.

In OART+Σ, first the set of stable features and their memory values are extracted from the training plans, and then the classification score is the sum of deviations from memory values of stable feature. Therefore, it provides a unique capability in interpretation of the classification results in terms of the input features. Not only can OART+Σ tell how much each feature (or feature category) of each plan is contributing to its classification score, but also it can suggest automatic improvements by providing a comparison of the stable long-term memory of the features with those of that plan.

For simplicity, we aggregated the features based on the three feature categories described in Sect 2 (ROI, dose, and machine features), but the methodology allows for individual feature analysis as well. Since the performance of $\text{OART}^{\tilde{U}} + \Sigma$ is better than $\text{OART}^{U} + \Sigma$, we show the results from $\tilde{U}$. The average deviations indicates the average feature-wise deviation from memory values:

$$\frac{1}{|\tilde{U}| \times |F_\sigma|} \sum_{\sigma \in \tilde{U}} \sum_{i \in F_\sigma} \Delta_\sigma(\mathbf{x})$$

Figs 9 and 10 show the breakdown of overall deviations according to feature categories.

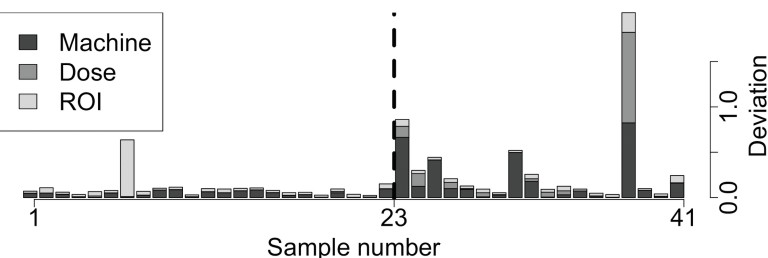

**Fig 9. Interpretable testing scores of OART for breast: The dashed line indicates the actual class separation, with unacceptable plans to the right.** Deviations are scaled.

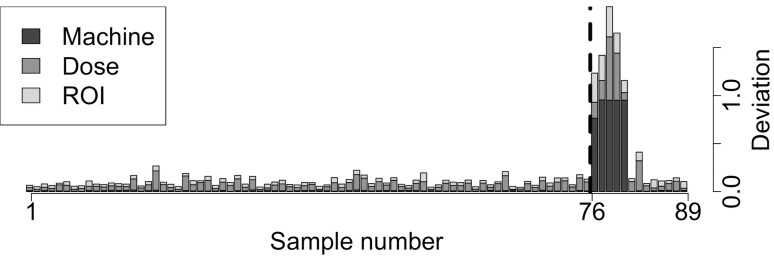

**Fig 10. Interpretable testing scores of OART for prostate: The dashed line indicates the actual class separation, with unacceptable plans to the right.** Deviations are scaled.

For example, the first five unacceptable testing plans of the prostate dataset (plan numbers 77 to 81) exhibit large deviations on the machine features. The next step for a clinician to improve the quality of the plan may then be to examine machine features for anomalies.

## 5 Discussion

As expected, one-class classifiers generally outperformed binary classifiers, and our OART approach was the best performer in all metrics for both the breast and prostate datasets. This performance is in spite of the fact that the one-class setting suffers from lower training/testing ratios, as mentioned in Sect 4, indicating that a one-class setting is the better anomaly detector. One previous binary classifier [2] used sophisticated class-imbalance mitigation techniques (namely, guided undersampling) to achieve better AUCs for these datasets, but it is theoretically limited in that it needs anomalies to learn, and may not perform as well in the presence of other types of unacceptable plans.

When OART is used as a classifier by summing deviations from stored memory values (OART+$\Sigma$), though it is slightly less accurate than OART+AE (Figs 7 and 8), it provides transparently interpretable scores, contrary to AE and OSVM (Figs 9 and 10). Moreover, although OART + $\Sigma$ commits to individual feature analysis for transparency, we included joint features (such as pairwise ROI and joint dose and shape features) in the original input features to accommodate some clinically-relevant mixed effects between features.

The difference in sparsity between the breast (more sparse) and prostate datasets shows that sparser datasets are more challenging in a one-class setting. Specifically, it becomes especially difficult to learn from $\approx 100$ samples with $\approx 1700$ features. A comparison of AUCs (Tables 2 and 3) clearly shows breast was a harder dataset supports this observation.

OART relies on distance parameter $\sigma$, but as a result of the one-class setting, it cannot be treated as a hyperparameter and tuned based on performance on the validation set. However, we did expect that OART benefit from upper-bounded distance parameter; As it would exclude classification scores coming from too high values of $\sigma$, where all features are stable enough to be included in the memory and practically no feature reduction is performed. The results showed that, especially in the case of OART + $\Sigma$, where no additional mixed feature classifier is applied, smaller $\sigma$ values indeed provide higher AUC. This observation provides additional insights into the interpretation of the distance parameter, and suggests that the accuracy may improve as a result of using variable $\sigma$ values for each feature.

Our results indicate that an automated QA can instantaneously accept up to 35% of breast and 81% of prostate plans, with no risk of accepting an unacceptable plan by mistake. Although implementation of such automated QA in a clinical setting requires further examination and compare between its accuracy and speed with those of human QA, our findings shed light on the sort of benefits that automation may provide through proper data storage and analytics.

## 6 Conclusions

We developed a feature extraction method, where features with stable values are selected into the memory, and the value of each selected feature is replaced by its deviation from the long-term memory of that feature. This method requires samples from only one class and is therefore suitable for such anomaly detection problems as quality assurance. Automated quality assurance, once developed to be reliable enough, offers greater speed and cost efficiency than human QA, saving valuable time for highly skilled personnel to focus on research.

The responsible use of AI necessitates the establishment of governance frameworks and compliance measures. Global AI policies and regulations are being developed, with a focus on sensitive areas such as healthcare[47]. Organizations are forming AI committees to guide decision-making and provide oversight. AI risk management frameworks help identify and mitigate potential risks throughout the AI lifecycle. Adhering to AI trust principles, such as integrity, explainability, fairness, and resilience, is crucial. Our explainable algorithm aligns with these governance frameworks and compliance requirements. Its transparency and interpretability enable stakeholders, including regulatory bodies, to understand and verify its decision-making process, fostering trust and supporting compliance with AI policies. By incorporating these frameworks and an explainable algorithm, our overall approach demonstrates a commitment to responsible and accountable AI deployment.

In this work, we considered deviations of equal magnitude as equally-significant evidence for anomalous behaviour. The future direction for this research may focus on modelling deviations probabilistically. Specifically, depending on the empirical distribution of feature values, an equal amount of deviation may not always imply an equal probability of an anomaly. A Bayesian approach, learning priors from historical data, can estimate the likelihood of a plan being acceptable or unacceptable.

The empirical results obtained from our study are promising, demonstrating the effectiveness of the our method from an empirical perspective. Future work may further validate the generalizability of these findings by applying the method to additional cancer types and data from various institutions. However, further theoretical analysis is needed to establish the convergence properties of our specific ART adaptation. While existing mathematical proofs demonstrate the convergence of ART networks to stable categories under certain conditions [48], additional theoretical work is needed to validate the convergence of our approach and applicability to other RT sites.

## Supporting information

**S1 Archive.** Archive.zip containing plotting data and code used to generate figures and results reported in the manuscript.
(ZIP)

## Author contributions

**Conceptualization:** Hootan Kamran.

**Data curation:** Hootan Kamran.

**Formal analysis:** Hootan Kamran.

**Funding acquisition:** Dionne Aleman, Chris McIntosh, Tom Purdie.

**Investigation:** Hootan Kamran.

**Methodology:** Hootan Kamran.

**Project administration:** Hootan Kamran.

**Resources:** Hootan Kamran.

**Software:** Hootan Kamran.

**Supervision:** Dionne Aleman, Chris McIntosh, Tom Purdie.

**Validation:** Hootan Kamran.

**Visualization:** Hootan Kamran.

**Writing – original draft:** Hootan Kamran.

**Writing – review & editing:** Hootan Kamran.

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
