## [Decision Letter · Decision Letter 0]

7 May 2024

PONE-D-24-09697Explainable one-class feature extraction by adaptive resonance for anomaly detection in quality assurancePLOS ONE

Dear Dr. Kamran,

Thank you for submitting your manuscript to PLOS ONE. After careful consideration, we feel that it has merit but does not fully meet PLOS ONE’s publication criteria as it currently stands. Therefore, we invite you to submit a revised version of the manuscript that addresses the points raised during the review process.

**Major revision:** * Please address the comments of the reviewers carefully in the revised submission. Thank you.

We look forward to receiving your revised manuscript.

Kind regards,

Zeyar Aung

Academic Editor

PLOS ONE

2. Please ensure you have stated the date of data collection in the Methods section of your manuscript text to fully comply with the PLOS ONE policy on reporting research involving human participants. This information is currently provided only in the Human Participants Research Checklist, which will not be published with your manuscript files.

Reviewers' comments:

Reviewer's Responses to Questions

**Comments to the Author**

1. Is the manuscript technically sound, and do the data support the conclusions?

Reviewer #1: Partly

Reviewer #2: Yes

2. Has the statistical analysis been performed appropriately and rigorously? 

Reviewer #1: Yes

Reviewer #2: No

3. Have the authors made all data underlying the findings in their manuscript fully available?

Reviewer #1: No

Reviewer #2: Yes

4. Is the manuscript presented in an intelligible fashion and written in standard English?

Reviewer #1: Yes

Reviewer #2: Yes

5. Review Comments to the Author

Reviewer #1: Looking into the manuscript, it is clear that the topic is timely and has potential implications to participants in the stock market. However, there are some significant issues that need to be addressed before the paper is considered for publication.

1. Introduction. The structure of the paper should be improved. In particular, the introduction section should be more structured, faster, and cover the key five elements: motivation, objectives, literature gap and paper's contributions to the literature, justification of the methodology, and summary of the results. Furthermore, in the introduction section, the authors clearly describe the added value of their analysis and the positioning in light of previous findings.

2. Related Work. I suggest the authors to analyze more papers that are recently published. Moreover, make sure that literature review is linked to a systematic.

3. The method used should be compared with various methods and the results should be strengthened. The methodological explanation of the method used should be made with the help of assumptions and formulas.

5. The results should be discussed more thoroughly in light of the existing literature. This is a very important point to address.

6. In the conclusion section, a more detailed discussion of the policy implications would make the paper richer and more informative for investors and policymakers.

Reviewer #2: Provide more examples or case studies to demonstrate the practical implementation of the proposed model in clinical settings. This would help in illustrating its applicability and effectiveness in real-world scenarios.

Expand the discussion on the limitations and challenges of your approach, especially focusing on situations where the model may face difficulties due to the unique characteristics of individual treatment plans or variations in equipment across different healthcare facilities.

Include a more detailed comparative analysis with other anomaly detection models that have been applied in similar healthcare settings. Highlighting the specific improvements your model offers over these existing methods would provide a clearer context for your contributions.

Discuss the model's adaptability to other types of quality assurance tasks outside of radiotherapy, potentially broadening the scope of your findings.

Enhance the explanation of how the adaptive resonance features work in layman's terms, to make the paper more accessible to readers who are not specialists in machine learning or medical physics.

Consider the ethical implications of automating such critical aspects of healthcare, particularly the reliance on algorithmic decisions in clinical environments. Discuss safeguards that could be implemented to prevent potential negative outcomes.

Propose future directions for this research, especially in terms of integrating this technology with other AI-driven tools used in healthcare, which could lead to a more comprehensive system for automated quality assurance across various treatment modalities.

Cite this https://doi.org/10.1109/ACCESS.2024.3373697 to improve your literature

6. PLOS authors have the option to publish the peer review history of their article (what does this mean?). If published, this will include your full peer review and any attached files.

Reviewer #1: **Yes: **Saad Mohamed Darwish

Reviewer #2: No

---

## [Author Response · Author response to Decision Letter 1]

7 Oct 2024

Reviewer 1

1. R1C1: Introduction. The structure of the paper should be improved. In particular, the

introduction section should be more structured, faster, and cover the key five elements: moti-

vation, objectives, literature gap and paper’s contributions to the literature, justification of the

methodology, and summary of the results. Furthermore, in the introduction section, the au-

thors clearly describe the added value of their analysis and the positioning in light of previous

findings.

We significantly re-wrote the Introduction section as you suggested and moved most of the

previous introduction to a new Related Work section. The Related Work section includes an

expanded literature review as requested in comments R1C1, R1C2, R1C4, and R2C3.

2. R1C2: Related Work. I suggest the authors to analyze more papers that are recently published.

Moreover, make sure that literature review is linked to a systematic.

We have updated the literature review with the following additional citations, which include

several papers from 2023 and 2024: Commission (2021); Gorokhov et al. (2017); Kehayias

et al. (2023); Li et al. (2023); Lundberg and Lee (2017); Ribeiro et al. (2016, 2018); Saab et al.

(2024); Salam et al. (2024); Union (2024)

3. R1C3: The method used should be compared with various methods and the results should be

strengthened. The methodological explanation of the method used should be made with the

help of assumptions and formulas.

We added a simple explanation of the main idea behind Adaptive Resonance Theory at the

beginning of the Methods section. Additionally, we extensively compare the various methods

and their results in the Discussion section.

4. R1C4: The results should be discussed more thoroughly in light of the existing literature. This

is a very important point to address.

We included four additional papers: Altman et al. (2015); Jensen et al. (2018); Kalendralis

et al. (2023); Kehayias et al. (2023). Our study stands out by using a comprehensive dataset

with over 100 variables and real erroneous plans, unlike previous works that focused on specific

error types, lower-dimensional data, or simulated errors. Our one-class approach overcomes

the limitations of requiring erroneous plans for training, providing a more robust and adapt-

able framework for detecting a wide range of unacceptable plans in radiation therapy. This

discussion is highlighted in blue on page 17 of the revised document.

5. R1C5: In the conclusion section, a more detailed discussion of the policy implications would

make the paper richer and more informative for investors and policymakers.

1

We added explanation on page 20, in accordance with the most prominent AI regulation

document, AI Act published by European Union Union (2024).

Reviewer 2

1. R2C1: Provide more examples or case studies to demonstrate the practical implementation of

the proposed model in clinical settings. This would help in illustrating its applicability and

effectiveness in real-world scenarios.

We added discussions about implementation and limitations in the Discussion section on page

20. Additional studies have been included in the discussion section (See R1C3 and R1C4 on

page 17), comparing their approaches, their results, and some of the implications of the clinical

implementation Kalendralis et al. (2023).

2. R2C2: Expand the discussion on the limitations and challenges of your approach, especially

focusing on situations where the model may face difficulties due to the unique characteristics

of individual treatment plans or variations in equipment across different healthcare facilities.

We added additional discussion of implementation challenges and limitations in the Discussion

section.

3. R2C3: Include a more detailed comparative analysis with other anomaly detection models that

have been applied in similar healthcare settings. Highlighting the specific improvements your

model offers over these existing methods would provide a clearer context for your contributions.

The Related Work section now includes a comparison with alternative methods, highlighting

that deep learning approaches often require additional algorithms for interpretation, lacking

inherent interpretability.

4. R2C4: Discuss the model’s adaptability to other types of quality assurance tasks outside of

radiotherapy, potentially broadening the scope of your findings.

We added a paragraph about other possible applications in the Discussion section.

5. R2C5: Enhance the explanation of how the adaptive resonance features work in layman’s terms,

to make the paper more accessible to readers who are not specialists in machine learning or

medical physics.

See R1C3.

6. R2C6: Consider the ethical implications of automating such critical aspects of healthcare,

particularly the reliance on algorithmic decisions in clinical environments. Discuss safeguards

that could be implemented to prevent potential negative outcomes.

See R1C5.

7. R2C7: Propose future directions for this research, especially in terms of integrating this tech-

nology with other AI-driven tools used in healthcare, which could lead to a more comprehensive

system for automated quality assurance across various treatment modalities.

We added a paragraph about future directions in the Conclusion section.

8. R2C8: Cite this https://doi.org/10.1109/ACCESS.2024.3373697 to improve your litera-

ture.

This paper (Salam et al., 2024) was cited as part of R2C4.

2

References

MB Altman, JA Kavanaugh, HO Wooten, OL Green, TA DeWees, H Gay, WL Thorstad, H Li, and

S Mutic. A framework for automated contour quality assurance in radiation therapy including

adaptive techniques. Physics in Medicine & Biology, 60(13):5199, 2015.

European Commission. Laying down harmonised rules on artificial intelligence (artificial intelligence

act) and amending certain union legislative acts. Eur Comm, 106:1–108, 2021.

Oleg Gorokhov, Mikhail Petrovskiy, and Igor Mashechkin. Convolutional neural networks for unsu-

pervised anomaly detection in text data. In International Conference on Intelligent Data Engi-

neering and Automated Learning, pages 500–507. Springer, 2017.

N Jensen, K Boye, S Damkjær, and I Wahlstedt. Impact of automation in external beam radiation

therapy treatment plan quality control on error rates and productivity. International Journal of

Radiation Oncology, Biology, Physics, 102(3):S149–S150, 2018.

Petros Kalendralis, Samuel MH Luk, Richard Canters, Denis Eyssen, Ana Vaniqui, Cecile Wolfs,

Lars Murrer, Wouter van Elmpt, Alan M Kalet, Andre Dekker, et al. Automatic quality assurance

of radiotherapy treatment plans using bayesian networks: A multi-institutional study. Frontiers

in oncology, 13:1099994, 2023.

CE Kehayias, D Bontempi, S Quirk, S Friesen, JS Bredfeldt, MA Huynh, H Aerts, RH Mak, and

CV Guthier. Deep learning-based automated quality assurance for palliative spinal treatment

planning in radiotherapy. International Journal of Radiation Oncology, Biology, Physics, 117(2):

S50, 2023.

Zhong Li, Yuxuan Zhu, and Matthijs Van Leeuwen. A survey on explainable anomaly detection.

ACM Transactions on Knowledge Discovery from Data, 18(1):1–54, 2023.

Scott M Lundberg and Su-In Lee. A unified approach to interpreting model predictions. Advances

in neural information processing systems, 30, 2017.

Marco Tulio Ribeiro, Sameer Singh, and Carlos Guestrin. ” why should i trust you?” explaining the

predictions of any classifier. In Proceedings of the 22nd ACM SIGKDD international conference

on knowledge discovery and data mining, pages 1135–1144, 2016.

Marco Tulio Ribeiro, Sameer Singh, and Carlos Guestrin. Anchors: High-precision model-agnostic

explanations. Proceedings of the AAAI Conference on Artificial Intelligence, 32(1), 2018.

Khaled Saab, Tao Tu, Wei-Hung Weng, Ryutaro Tanno, David Stutz, Ellery Wulczyn, Fan Zhang,

Tim Strother, Chunjong Park, Elahe Vedadi, Juanma Zambrano Chaves, Szu-Yeu Hu, Mike

Schaekermann, Aishwarya Kamath, Yong Cheng, David G. T. Barrett, Cathy Cheung, Basil

Mustafa, Anil Palepu, Daniel McDuff, Le Hou, Tomer Golany, Luyang Liu, Jean baptiste Alayrac,

Neil Houlsby, Nenad Tomasev, Jan Freyberg, Charles Lau, Jonas Kemp, Jeremy Lai, Shekoofeh

Azizi, Kimberly Kanada, SiWai Man, Kavita Kulkarni, Ruoxi Sun, Siamak Shakeri, Luheng He,

Ben Caine, Albert Webson, Natasha Latysheva, Melvin Johnson, Philip Mansfield, Jian Lu, Ehud

Rivlin, Jesper Anderson, Bradley Green, Renee Wong, Jonathan Krause, Jonathon Shlens, Ewa

Dominowska, S. M. Ali Eslami, Katherine Chou, Claire Cui, Oriol Vinyals, Koray Kavukcuoglu,

James Manyika, Jeff Dean, Demis Hassabis, Yossi Matias, Dale Webster, Joelle Barral, Greg Cor-

rado, Christopher Semturs, S. Sara Mahdavi, Juraj Gottweis, Alan Karthikesalingam, and Vivek

Natarajan. Capabilities of gemini models in medicine, 2024.

3

Abdu Salam, Mohammad Abrar, Farhan Amin, Faizan Ullah, Izaz Ahmad Khan, Bader Fahad

Alkhamees, and Hussain AlSalman. Securing smart manufacturing by integrating anomaly de-

tection with zero-knowledge proofs. IEEE Access, 12:36346–36360, 2024. doi: 10.1109/ACCESS.

2024.3373697.

European Union. Artificial intelligence act. european parliamentary research service, 2023, 2024.

---

## [Decision Letter · Decision Letter 1]

1 Dec 2024

PONE-D-24-09697R1Explainable one-class feature extraction by adaptive resonance for anomaly detection in quality assurancePLOS ONE

Dear Dr. Kamran,

Thank you for submitting your manuscript to PLOS ONE. After careful consideration, we feel that it has merit but does not fully meet PLOS ONE’s publication criteria as it currently stands. Therefore, we invite you to submit a revised version of the manuscript that addresses the points raised during the review process.

**Minor revision:  Some more revisions are still required. Please address the comments by the reviewers.**

We look forward to receiving your revised manuscript.

Kind regards,

Zeyar Aung

Academic Editor

PLOS ONE

**Journal Requirements:**

Reviewers' comments:

Reviewer's Responses to Questions

**Comments to the Author**

1. If the authors have adequately addressed your comments raised in a previous round of review and you feel that this manuscript is now acceptable for publication, you may indicate that here to bypass the “Comments to the Author” section, enter your conflict of interest statement in the “Confidential to Editor” section, and submit your "Accept" recommendation.

Reviewer #2: All comments have been addressed

Reviewer #3: (No Response)

2. Is the manuscript technically sound, and do the data support the conclusions?

Reviewer #2: Yes

Reviewer #3: Yes

3. Has the statistical analysis been performed appropriately and rigorously? 

Reviewer #2: I Don't Know

Reviewer #3: Yes

4. Have the authors made all data underlying the findings in their manuscript fully available?

Reviewer #2: Yes

Reviewer #3: No

5. Is the manuscript presented in an intelligible fashion and written in standard English?

Reviewer #2: Yes

Reviewer #3: Yes

6. Review Comments to the Author

**Reviewer #2:** The paper needs to check for spelling, typo and grammatical mistakes. The rest of the paper is now ok.

**Reviewer #3:** 1. Although additional citations were included, the review could benefit from a systematic and thematic organization. There are still areas where the links between referenced studies and the proposed method remain unclear.

2. The inclusion of comparative analysis with alternative methods is appreciated, but the discussion is surface-level. A deeper dive into why the proposed method outperforms others with specific examples would be more impactful.

3. Figures and Tables - Ensure clarity and alignment with the narrative.

7. PLOS authors have the option to publish the peer review history of their article (what does this mean?). If published, this will include your full peer review and any attached files.

Reviewer #2: No

Reviewer #3: No

---

## [Author Response · Author response to Decision Letter 2]

6 Feb 2025

\section*{Journal Requirements:}

``Please review your reference list to ensure that it is complete and correct.''

\textcolor{darkgreen}{We have reviewed the reference list to ensure that it is complete and correct. We have checked all 49 citations, and none of them have been retracted.}

\section*{Response to reviewers}

We have carefully revised our manuscript, "Explainable one-class feature extraction by adaptive resonance for anomaly detection in quality assurance" (\# PONE-D-24-09697), based on the insightful feedback provided by the editor and reviewers. This letter details our point-by-point responses to each comment, clarifying how we have incorporated the suggestions into the revised manuscript.

For ease of reference

\begin{itemize}

\item Reviewers' comments: Black text

\item Our responses: Dark green text

\item Changes in the manuscript: Marked in blue and tagged with the corresponding comment number (e.g., R2C1 for Reviewer 2, Comment 1).

\end{itemize}

The changes include spelling and grammatical corrections to address Reviewer 2's comments and revisions to the Related Work section based on Reviewer 3's feedback.

\subsection*{Reviewer 2}

Reviewer \#2: The paper needs to check for spelling, typo and grammatical mistakes. The rest of the paper is now ok.

\textcolor{darkgreen}{We appreciate Reviewer 2's careful reading and helpful suggestions for improving the clarity and correctness of our manuscript. We have made the following changes, marked in blue with the code R2C1 in the manuscript:}

\textcolor{darkgreen}{

\begin{itemize}

\item \textbf{In the Abstract:}

\begin{itemize}

\item Improved the flow of the sentence beginning with "This issue" to enhance readability.

\end{itemize}

\item \textbf{In Related Work:}

\begin{itemize}

\item Corrected the spelling of "occurred" (previously "occured").

\item Ensured consistent use of "a small number of plans" throughout.

\item Removed the hyphen in "label imbalanced" for better readability.

\item Changed "in favour of" to "in favor of" to maintain consistency with American English spelling.

\end{itemize}

\item \textbf{In Section 2:}

\begin{itemize}

\item Changed "OSVM" to "one-class SVMs" to maintain consistency with the use of spelled-out forms.

\end{itemize}

\item \textbf{In Section 4:}

\begin{itemize}

\item Removed the redundant word "hyperreferencced."

\item Corrected the capitalization of "Breast dataset" and "Prostate dataset."

\item Corrected the spelling of "features" (previously "featues").

\item Rephrased the sentence about the stopping condition for training iterations for clarity.

\item Rephrased the sentence about the two cases for conciseness.

\item Maintained consistent use of either spelled-out forms or abbreviations for "PCA," "ICA," and "autoencoders."

\item Split the sentence about PCA and ICA components for clarity.

\item Maintained consistent use of the spelled-out form for "autoencoder."

\item Replaced "each with" with "using" for better flow.

\item Replaced "smallest" with "lowest" for better word choice.

\item Rephrased the sentence about the classification based on OART's feature values for clarity.

\end{itemize}

\item \textbf{In Section 5:}

\begin{itemize}

\item Changed "in general" to "generally" for better grammar.

\item Replaced "demonstrates" with "shows" for better word choice.

\item Rephrased the sentence about the hyperparameter for clarity.

\item Replaced "do in fact" with "indeed" for better flow.

\item Rephrased the sentence about automated QA for clarity.

\item Rephrased the sentence about governance frameworks for clarity.

\end{itemize}

\item \textbf{In the Conclusions:}

\begin{itemize}

\item Rephrased the sentence about the speed and cost-efficiency of automated QA for clarity and impact.

\item Rephrased the sentence about future research directions for conciseness.

\item Rephrased the sentence about Bayesian approaches for clarity.

\item Rephrased the sentence about validating the generalizability of findings for clarity.

\item Rephrased the sentence about the need for further theoretical analysis for conciseness.

\item Rephrased the sentence about existing mathematical proofs for clarity.

\item Rephrased the sentence about future research addressing convergence for conciseness.

\item Rephrased the sentence about combining empirical evidence and theoretical foundations for impact.

\end{itemize}

\end{itemize}}

\subsection*{Reviewer 3}

\begin{enumerate}

\item{{\textbf{R3C1}} Although additional citations were included, the review could benefit from a systematic and thematic organization. There are still areas where the links between referenced studies and the proposed method remain unclear. The inclusion of comparative analysis with alternative methods is appreciated, but the discussion is surface-level. A deeper dive into why the proposed method outperforms others with specific examples would be more impactful.}

\textcolor{darkgreen}{We appreciate Reviewer 3's careful reading and helpful suggestions for improving the clarity and correctness of our manuscript. We have made the following changes, marked in blue with the code {\textbf{R3C1}} in the manuscript:}

\textcolor{darkgreen}{We have revised multiple parts of the manuscript to improve the connection between previous studies and our proposed method:

\begin{itemize}

\item {We clarified why handling sparsity in high-dimensional feature spaces is a key challenge in RT planning and explicitly referenced prior work addressing this issue. }

\item {We strengthened the explanation of why binary classification methods are insufficient for RT QA, emphasizing the risk of failing to detect novel anomalies. }

\item {We improved the description of how our OART method incorporates feature selection to enhance anomaly detection while maintaining interpretability.}

\end{itemize}

}

\item {{\textbf{R3C2}} Figures and Tables - Ensure clarity and alignment with the narrative.}

\end{enumerate}

\textcolor{darkgreen}{We have carefully reviewed all figures and tables in the manuscript to ensure their clarity and alignment with the surrounding narrative. This review included checks for proper referencing, clear descriptions, high-resolution images, and accurate labels and explanations. If there are any additional specific issues with the tables and figures, please let us know the details so we can address them effectively.

}

---

## [Editor Report · Decision Letter 2]

14 Mar 2025

Explainable one-class feature extraction by adaptive resonance for anomaly detection in quality assurance

PONE-D-24-09697R2

Dear Dr. Kamran,

We’re pleased to inform you that your manuscript has been judged scientifically suitable for publication and will be formally accepted for publication once it meets all outstanding technical requirements.

Kind regards,

Zeyar Aung

Academic Editor

PLOS ONE

Additional Editor Comments (optional):

The paper is acceptable now. Please remove the tags R3C1, etc. from the final submission. Thank you.
---

## [Editor Report · Acceptance letter]

PONE-D-24-09697R2

PLOS ONE

Dear Dr. Kamran,

I'm pleased to inform you that your manuscript has been deemed suitable for publication in PLOS ONE. Congratulations! Your manuscript is now being handed over to our production team.

Kind regards,

on behalf of

Dr. Zeyar Aung

Academic Editor

PLOS ONE